



# Transient climate simulations of the Holocene (version 1) – experimental design and boundary conditions

Zhiping Tian[1], Dabang Jiang[1, 2], Ran Zhang[1], Baohuang Su[3]

[1]Institute of Atmospheric Physics, Chinese Academy of Sciences, Beijing 100029, China
[2]Collaborative Innovation Center on Forecast and Evaluation of Meteorological Disasters, Nanjing University of Information Science & Technology, Nanjing, 210044, China
[3]State Key Laboratory of Severe Weather, Chinese Academy of Meteorological Sciences, China Meteorological Administration, Beijing 100081, China

*Correspondence to*: Zhiping Tian (tianzhiping@mail.iap.ac.cn)

**Abstract.** The Holocene, started approximately 11.5 thousand years before present (ka), is the latest interglacial period with several rapid climate changes from decades to centuries time scales superimposed on the millennium scale mean climate trend. Climate models provide useful tools to investigate the underlying dynamic mechanisms for the climate change during this well-studied time period. Thanks to the improvements of the climate model and computational power, transient simulation of
the Holocene offers an opportunity to investigate the climate evolution in response to time-varying external forcings and feedbacks. Here, we present the design of a new set of transient experiments for the whole Holocene from 11.5 ka to the preindustrial period (1850 CE) (HT-11.5ka) to investigate both combined and separated effects of the main external forcing of orbital insolation, atmospheric greenhouse gas (GHG) concentrations, and ice sheets on the climate evolution over the Holocene. The HT-11.5ka simulations are performed with a relatively high-resolution version of the comprehensive Earth
system model CESM1.2.1 without acceleration, both fully- and singly-forced by time-varying boundary conditions of orbital configurations, atmospheric GHGs, and ice sheets. Preliminary simulation results show a slight decrease of the global annual mean surface air temperature from 11.5 ka to 7.5 ka due to both decreases in orbital insolation and GHG concentrations, with an abrupt cooling at approximate 7.5 ka, which is followed by a continuous warming until the preindustrial period mainly due to increased GHG concentrations. The simulated cooling magnitude at 6 ka lies within the range of the 14 PMIP4 model results
and is close to their median result for the mid-Holocene simulations. Further analyses on the HT-11.5ka transient simulation results will be covered by follow-up studies.

## 1 Introduction

### 1.1 Climate evolution over the Holocene

Since the end of the Younger Dryas cooling event at ~11.5 thousand years before present (ka), the global climate has entered
the Holocene period. The Holocene is the latest interglacial period in the Quaternary glacial–interglacial cycles, which has

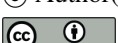



experienced strong global environmental changes mainly due to the retreat of continental ice sheets, changes in atmospheric greenhouse gas (GHG) concentrations, and variations in the seasonal insolation associated with changes in the Earth's orbital parameters and solar variability (Mayewski et al., 2004). Although previous attempts to subdivide the Holocene still remain inconclusive, the Holocene can roughly be divided into three periods (Nesje and Dahl, 1993): the early Holocene with a high

boreal summer insolation and a large remnant ice sheet in North America, the Holocene Hypsithermal with a continuing high boreal summer insolation but a small remnant North American ice sheet, and the late Holocene Neoglacial with a declining boreal summer insolation (Wanner and Ritz, 2011; Wanner et al., 2011; Walker et al., 2012).

The early Holocene, approximately spanning from 11.5 to 9 ka, represents the last transition from full glacial to interglacial conditions, which is characterized by a warming trend in the Northern Hemisphere, mainly in the mid- and high-latitudes, as

revealed from numerous reconstructions (Hoelzmann et al., 2004; Shakun et al., 2012). Although with differences in magnitude, the early Holocene warming was consistently indicated from ice cores in Greenland (Grootes et al., 1993; Vinther et al., 2008) and the Canadian High Arctic (Koerner and Fisher, 1990), lake sediments in western and northern Europe (Brooks and Birks, 2000; Brooks et al., 2012; Birks, 2015), speleothems in China (Yuan et al., 2004; Wang et al., 2005), and marine sediments in the North Atlantic (Bond et al., 1993; Kandiano et al., 2004; Hald et al., 2007). In tropical regions, the change in precipitation

was more prominent during the early Holocene, among which the African Humid Period is one typical example (Gasse, 2001; Hoelzmann et al., 2004). Proxy-based evidence also shows a humidity climate during this transition period from lake sediments in the Arabian Sea and the Bay of Bengal in coincidence with the stronger Asian southwest monsoon (Gupta et al., 2003; Staubwasser, 2006) and from the marine sediment off the Venezuelan coast in the northern South America in relation to the southward migration of the Intertropical Convergence Zone (Haug et al., 2001), which was suggested to be fundamentally

affected by precession (e.g., Renssen et al., 2004). In contrast, a drier climate was inferred from the paleoenvironmental and archaeological evidence in the central and southern South America (Maslin and Burns, 2000; Seltzer et al., 2000; Núñez et al., 2002). In addition, reconstructions from a variety of archives have shown that the westerlies-dominated North Asia was markedly drier than today during the early Holocene, whereas the monsoon-influenced East Asia was substantially wetter but with asynchronous peak of precipitation in Northeast, Northwest, and North China (An et al., 2000; Chen et al., 2008, 2019;

Zhang et al., 2011).

The Holocene Hypsithermal, covering from about 9 ka to 5–6 ka, generally refers to the interval of warmth associated with the peak Holocene temperature, which is also called the mid-Holocene, the Holocene Thermal Maximum, the Holocene Climatic Optimum, Altithermal or Megathermal (Deevey and Flint, 1957; Wanner et al., 2008; Ljungqvist, 2011). The timing and spatial pattern of this period in different regions have long attracted the attention of researchers. For example, the Holocene

megathermal period was defined as 8.5–3.0 ka in China by Shi et al. (1994) according to multiple evidence from pollen, ice core, paleo-lake, and paleosol data. Relative to the preindustrial period, the reconstructed Holocene maximum temperature appeared during 10–8 ka in the North Atlantic and the adjacent polar regions, between early and middle Holocene at the northern mid- and high-latitudes, during 7–5 ka in northern Europe and northwestern North America, suggesting that these local warm periods were very likely not globally synchronous (Jansen et al., 2007). Based on 140 sites across the Western



Hemisphere of the Arctic, the Holocene Thermal Maximum was indicated to be time-transgressive with an earlier start and end in western Arctic and a later start and end in the eastern part, especially in the southeastern part, due to the effect of the Laurentide ice sheet melting (Kaufman et al., 2004). Different from the Northern Hemisphere, the southern mid- and high-latitudes experienced a declining of temperature during the middle and late Holocene following the early Holocene warming period (Shevenell et al., 2011). Concerning the moisture conditions, paleoenvironmental data from lake level, vegetation and

isotope records, and aeolian deposits consistently showed that the midcontinental extra-tropics were wetter with cooler summers than today during the mid-Holocene (Harrison et al., 2015).

The late Holocene Neoglacial or Neoglaciation, lasting from about 5–6 ka to preindustrial period, is featured by an obvious increase in the number of glacial advances worldwide mainly due to decreasing solar forcing in the boreal summer (Solomina et al., 2015). The global mean surface temperature during this period displayed a consistent cooling trend as indicated from

multi-proxy-based reconstructions from 11 selected temperature time series (Wanner et al., 2008), 37 terrestrial and 23 marine temperature records (Ljungqvist, 2011), 73 globally distributed temperature records (Marcott et al., 2013), and a global compilation of temperature time series from 679 sites (Kaufman et al., 2020a, b) and 1302 records (Zhang et al., 2022). This late Holocene cooling trend was suggested to be mainly due to the influence of orbital forcing together with the melting ice sheet in North America. However, as pointed out by Wanner (2021), most of those studies are mainly based on marine records

which have seasonal (mostly summer) biases in many cases, and boreal winter season, when solar insolation was increasing in the tropics and Southern Hemisphere, is only weakly represented by proxies. In fact, after removing the seasonal bias in the marine record located in the tropical western Pacific, the annual mean sea surface temperature was steadily increasing since about 12 ka (Bova et al., 2021). The late Holocene warming trend was further supported by simulations (e.g., Liu et al., 2014; Liu et al., 2018) and by several reconstructions from subfossil pollen at 642 sites across North America and Europe (Marsicek

et al., 2018) and from two stalagmites records in the southern Ural Mountains over western continental Eurasia during the winter season (Baker et al., 2017), which was mainly attributed to rising atmospheric GHGs and winter insolation. Therefore, it still remains inconclusive on whether the late Holocene was cooler or warmer. To reconcile this so called "Holocene temperature conundrum", further efforts should be taken to the improvements both in simulations, by including solar irradiance forcing, land-use changes, dust effects, and stratospheric chemistry and dynamics, and in reconstructions with higher spatial

and temporal resolutions, a balanced distribution between marine and terrestrial and between the Northern and Southern Hemisphere data, and the representing proxies from both summer and winter seasons (Wanner, 2021; Zhang et al., 2022), as well as in the model–data assimilation approaches (Osman et al., 2021).

Superimposed on the millennium scale mean climate trend during the above Holocene periods are several rapid cooling events on the time scales from decades to centuries (Wanner et al., 2011; Walker et al., 2012). Based on ~50 globally distributed

paleoclimate records, six periods of significant rapid climate changes were identified during 9–8 ka, 6–5 ka, 4.2–3.8 ka, 3.5–2.5 ka, 1.2–1.0 ka, and 0.6–0.15 ka by Mayewski et al. (2004), which generally cover the rapid cold events occurring at about 8.2 ka, 5.5 ka, 4.2 ka, 2.8 ka, 1.4 ka, and 0.4 ka ("Little Ice Age") according to studies of ice rafted debris in the North Atlantic (Bond et al., 1997; Moros et al., 2004). Most of those events are characterized by cooling at northern mid- and high-latitudes,



aridity at low-latitude monsoon regions, and major atmospheric circulation changes (Mayewski et al., 2004). The cold events

during the early Holocene before 8.2 ka were generally inferred to relate to the freshwater outbreak caused by ice melting in the North Atlantic, while those during the mid- and late Holocene may mainly associate with the solar variability superimposed on long-term changes in insolation; explosive volcanic eruptions and fluctuations of the thermohaline circulation may also play important roles (Mayewski et al., 2004; Magny et al., 2007; Wanner et al., 2011).

## 1.2 Transient modelling of the Holocene

Transient modelling of the Holocene offers an opportunity to examine the climate evolution in response to time-varying external forcings and feedbacks. Combined with proxy data, it is the most useful tool to comprehensively understand the causes for past complex and interrelating climate processes. Since the 1990s, the Paleoclimate Modelling Intercomparison Project (PMIP) has served to coordinate paleoclimate simulations and model–data comparisons (Joussaume and Taylor, 1995; Braconnot et al., 2007; Taylor et al., 2012), and it now enters its fourth phase (PMIP4; Kageyama et al., 2018) contributing to

the current sixth phase of the Coupled Model Intercomparison Project (CMIP6; Eyring et al., 2016) and the Intergovernmental Panel on Climate Change (IPCC) Sixth Assessment Report (IPCC, 2021). The major focuses in PMIP4 include not only equilibrium (or time-slice) simulations to investigate the impact of changes mainly in orbital forcing, e.g., during the mid-Holocene (6 ka), but also transient simulations of the last deglaciation from 21 to 9 ka with time-varying orbital forcing, GHGs, ice sheets, and other geographical changes (Ivanovic et al., 2016) and of the Holocene from 6 to 0 ka with temporal changes

in orbital forcing and GHGs (Otto-Bliesner et al., 2017).

Up to now, numerous transient simulations covering the Holocene have been carried out using global climate models with different complexities. For example, FOAM, a fully coupled global atmosphere–ocean general circulation model with a low resolution of 4° × 7.5° (latitude × longitude), was used to run a transient simulation for the past 280 ka to examine the evolutionary response of global monsoons to orbital forcing, which is accelerated by a factor of 100 (Kutzbach et al., 2008).

FAMOUS (Smith et al., 2008), a low-resolution (5° × 7.5°) version of the HadCM3 (Gordon et al., 2000) coupled model, was used to conduct a number of accelerated transient experiments (by a factor of 10) with single and full forcings of orbital parameters, GHGs, and ice sheets for the last 120 ka to investigate the physical climate of the atmosphere and ocean through the last glacial cycle (Smith and Gregory, 2012). A set of transient simulations of the last deglaciation for the past 26 ka was performed with a low resolution (~3.75° × 3.75°) Earth system model MPI-ESM1.2 to explore differences in climate response

to different underlying ice-sheet reconstructions and methods of meltwater distribution (Kapsch et al., 2020) as suggested within the PMIP4 deglaciation protocol (Ivanovic et al., 2016). The Transient Climate Evolution of the past 21 ka (TraCE-21ka; He, 2011), run with the low-resolution (~3.75° × 3.75°) National Center for Atmospheric Research (NCAR) Community Climate System Model version 3 (CCSM3), is widely used for examining the climate evolution since the last glacial maximum at global and regional scales (e.g., Liu et al., 2014; Liu et al., 2021; Wu et al., 2021). A fully coupled atmosphere–ocean–sea

ice model ECBilt-CLIO, with a horizontal resolution of ~5.6° × 5.5°, was also used for transient simulations of the last deglaciation (last 21 ka) to investigate the effect of using accelerated (by a factor of 10) boundary conditions and of



uncertainties in the initial state (Timm and Timmermann, 2007); additionally coupled with a vegetation model, ECBilt-CLIO-VECODE3 was performed to investigate the global characterization of the Holocene thermal maximum for the last 9 ka (Renssen et al., 2012); a later developed version LOVECLIM, additionally coupled with carbon cycle and terrestrial ice sheets

(Goosse et al., 2010), was employed to examine the effects of melting ice sheets and orbital forcing on the early Holocene (11.5–7 ka) warming (Zhang et al., 2016) and the obliquity and $CO_2$ effects on southern climate during the past 408 ka with an acceleration factor of 5 (Timmermann et al., 2014). The three Holocene transient simulations performed with the LOVECLIM, CCSM3, and FAMOUS were compared to explore the Holocene temperature trends in the extratropical Northern Hemisphere and the spatial contrasts of the Holocene hydroclimate trend between North and East Asia (Zhang et al., 2018,

140 2020).

More concentrate on the Holocene, NNU-Hol (Nanjing Normal University-Holocene) transient simulations for the past 12 ka, run with an acceleration factor of 10 with the low-resolution (~3.75° × 3.75°) Community Earth System Model (CESM) version 1.0.3 (CESM1.0.3), successfully reconciled the temperature conundrum with a cooling trend during 5–0.15 ka by considering the volcanic forcing (Wan et al., 2020); they also showed an increase trend of the northern mid-latitude

precipitation during 7–0 ka in response to the orbital forcing, which is consistent with the reconstructions although the trend is concentrated in marine areas (Sun et al., 2020). A transient simulation of the last 8 ka performed with the high-resolution (1.875° × 1.875°) MPI-ESM1.2, additionally forced by solar variability and volcanic forcing, further revealed that both a global warming and cooling modes coexisted during the Holocene, with the former mode dominating in the mid-Holocene and the latter taking over in the late Holocene, and thus shed light on the Holocene temperature conundrum (Bader et al., 2020). The

MPI-ESM1.2 Holocene transient simulation also captured the global vegetation transitions and the time-transgressive end of the African humid period as revealed in proxy data, with an earlier end of the African humid period in the north/east than in the south/west due to the regionally varying dynamical controls on precipitation (Dallmeyer et al., 2020, 2021). A transient climate simulation of the past 9.5 ka, conducted with a low resolution (~3.75° × 3.75°) atmosphere–ocean–sea ice coupled model KCM with an acceleration factor of 10, exhibited a wave train spatial pattern and diverging trends of summer

precipitation across the Asian monsoon region in response to the orbital forcing (Jin et al., 2014). Two versions of the IPSL model, i.e., the low-resolution (1.875° × 3.75°) version of IPSLCM5A-LR and modified and medium-resolution (1.25° × 2.5°) of IPSLCM5A-MR with a new hydrological model, a prognostic snow model, and a dynamical vegetation module, were used to run the transient simulations of the last 6 ka, indicating that different model versions and experimental setups have a larger impact on the mean state of climate, e.g., the drying trend of the Indian and West African summer monsoon rainfall, than on

the associated interannual-to-decadal variability and the vegetation–climate interactions (Braconnot et al., 2019a, b; Crétat et al., 2020). In addition, the four Holocene transient simulations run with MPI-ESM1.2, two versions of the IPSL model, and AWI-ESM2 (1.875° × 1.875°; Sidorenko et al., 2019) were compared to support the orbital forcing of ENSO amplitude since the mid-Holocene (Carré et al., 2021) and highlight the importance of the application of calendar transformation in the analysis of climate simulations when we do multi-model comparisons (Shi et al., 2021).





### 1.3 A new set of Holocene transient experiments


Taken together, among the numerous sets of Holocene transient simulations published in the past, only seven of them covered the whole Holocene of past 11.5 ka, i.e., simulations with LOVECLIM for past 408 ka, FOAM for past 284 ka, FAMOUS for past 120 ka, MPI-ESM1.2 for past 26 ka, CCSM3 and ECBilt-CLIO for past 21 ka, and CESM1.0.3 for past 12 ka. They were performed with climate system models or Earth system models with intermediate complexity, all of which have relatively low

spatial resolutions varying from ~5.6° × 5.5° to ~3.75° × 3.75° and/or use acceleration techniques by a factor from 5 to 100 to reduce computational resources. However, the model resolution determines the level of detail in its representation of physical processes related with the topography, heat transport, sea ice cover, etc., which may have large effects on the Holocene climate evolution (Zhang et al., 2018). On the other hand, acceleration in transient simulations can also have a significant impact on the local climate history, for example, which will lead to damped and delayed temperature response to the boundary conditions

in the North Atlantic (Timm and Timmermann, 2007). In addition, most models used for the Holocene transient simulations are fully-forced by time-varying boundary conditions of orbital configurations, GHGs, and ice sheets, some of which are additionally forced by meltwater flux, land use/land cover, solar variability, or volcanic forcing, making it difficult to isolate their relative contribution to the past climate evolution.

In this context, it is urgent to carry out a new set of unaccelerated transient experiments focusing on the whole Holocene since

11.5 ka using the comprehensive Earth system model with relatively high spatial resolutions, considering both the full-forcing and single-forcing effects of several most important boundary conditions, including the orbital parameters, GHG concentrations, and ice sheets, on the Holocene climate evolution. These set of Holocene transient simulations for the last 11.5 ka will also fill in the gap between the PMIP4 transient Core experiments for the last deglaciation from 21 to 9 ka (Ivanovic et al., 2016) and the PMIP4 Tier 3 transient simulations for the Holocene from 6 to 0 ka (Otto-Bliesner et al., 2017).

The aim of this paper is to outline the model description (in Section 2), experimental setup, and boundary conditions (in Section 3) for the new Holocene transient climate experiments of the last 11.5 ka (henceforth HT-11.5ka for brevity), with some preliminary results being provided in Section 4. Further analyses on these simulation results will be covered by follow-up studies.

### 2 Model description

The model used to run the HT-11.5ka simulation in this study is the NCAR CESM version 1.2.1 (CESM1.2.1), which was released on December 2013 and was a relatively new version when we started this simulation on 2016. CESM is a fully-coupled global Earth system model that provides state-of-the-art simulations of the Earth's past, present, and future climate states, with an earlier version 1 (CESM1) participated in the CMIP phase 5 (CMIP5; Taylor et al., 2012) and the later version 2 (CESM2) participated in the CMIP phase 6 (CMIP6; Eyring et al., 2016). CESM1.2.1 consists of four components and can

be coupled in different configurations. The atmospheric component we used here is the NCAR Community Atmosphere Model version 4 (CAM4; Neale et al., 2010), also used in the CCSM version 4 (CCSM4; Gent et al., 2011), which employs a finite





volume dynamical core at a horizontal resolution of approximately $1.9° \times 2.5°$ (latitude $96 \times$ longitude 144 grid points) with 26 vertical layers. The land component is the Community Land Model version 4.0 (CLM4; Lawrence et al., 2011) with the carbon nitrogen–dynamic global vegetation model turned off, which is run on the same horizontal grid as the atmospheric

component. The oceanic component is the Parallel Ocean Program version 2 (POP2; Danabasoglu et al., 2012) based on the POP version 2.1 of the Los Alamos National Laboratory (Smith et al., 2010), which uses an approximately 1° horizontal grid with 60 vertical levels. The sea ice component is the Los Alamos Sea Ice Model (or referred to as the Community Ice Code) version 4.0 (CICE 4.0; Hunke and Lipscomb, 2008) with the same horizontal grid as the ocean component. These four components are coupled by the version 7 coupler (CPL7) (Craig et al., 2012). More details on the model are available online

at https://www.cesm.ucar.edu/models/cesm1.2/ and in Hurrell et al. (2013), and further information on the model simulations for the past and present climates can be found in Park et al. (2019) and Song and Zhang (2018, 2019).

### 3 Experimental designs for the HT-11.5ka simulations

The HT-11.5ka simulations of the Holocene focus on the period from 11.5 ka to the preindustrial period (1850 CE), which will be the supplement for the PMIP4 Core simulations of the last deglaciation (version 1) spanning from 21 to 9 ka (Ivanovic et

al., 2016). Note that 0 ka is 1950 CE, and 1850 CE is equivalent to 0.1 ka. All simulations spin up from the early Holocene at 11.5 ka and all boundary conditions are available from 11.5 ka to the preindustrial period. The initialization state at 11.5 ka and boundary conditions since 11.5 ka, including Earth's orbital parameters, atmospheric GHG concentrations, and ice sheets and orography, are summarized in Table 1 and described below. Boundary condition data for the HT-11.5ka transient experiments are provided on the PMIP4 wiki for the transient deglaciation (PMIP Last Deglaciation Working Group, 2016;

https://pmip4.lsce.ipsl.fr/doku.php/exp_design:degla).

### 3.1 Early Holocene spin-up

The early Holocene spin-up experiment for the HT-11.5ka simulations is an equilibrium or time slice simulation with boundary conditions of orbital parameters, GHGs, and ice sheets prescribed at 11.5 ka (Table 1). Orbital parameters of eccentricity, obliquity, and longitude of perihelion is set to 0.01957212, 24.17958°, and 270.2132°, respectively, with the date of vernal

equinox fixed at the noon on 21 March. The solar constant is fixed to 1365 W m$^{-2}$, which is consistent with the preindustrial value for the PMIP phase 3 (PMIP3) and CMIP phase 5 (CMIP5) experiments but higher than the value recommended for the transient deglaciation simulation (e.g., $1361.0 \pm 0.5$ W m$^{-2}$; Mamajek et al., 2015) and the PMIP4/CMIP6 experiments (e.g., $1360.8 \pm 0.5$ W m$^{-2}$; Matthes et al., 2017; Otto-Bliesner et al., 2017). The atmospheric GHGs is prescribed to 267.4 ppm for carbon dioxide ($CO_2$), 652.4 ppb for methane ($CH_4$), 251.1 ppb for nitrous oxide ($N_2O$), 0 for chlorofluorocarbons (CFCs),

and the PMIP3/CMIP5 preindustrial (1850 CE) value for ozone ($O_3$). These values are compatible with the time-evolving boundary conditions for the HT-11.5ka simulations and consistent with the latest ice-core age model (AICC2012; Veres et al., 2013) and records (Loulergue et al., 2008; Schilt et al., 2010; Bereiter et al., 2015). Prescribed ice sheets and the associated



topography use the 11.5 ka data from the ICE-6G_C (VM5a) reconstruction (henceforth ICE-6G_C; Argus et al., 2014; Peltier et al., 2015). All other boundary conditions, such as the coastlines, bathymetry, vegetation and land cover, and aerosols, are

prescribed at the preindustrial levels.

Forced by the above boundary conditions, the early Holocene spin-up is continuously integrated for 1500 years, with the absolute value of the trend in global mean surface air and sea surface temperatures being less than 0.05 °C and the Atlantic Meridional Overturning Circulation being stable for the last 100 years, which means the model has reached a quasi-equilibrium state as suggested by Kageyama et al. (2018). All Holocene transient experiments start from this equilibrium state at 11.5 ka

and run transiently until the preindustrial period at 1850 CE.

### 3.2 Holocene transient simulations

### 3.2.1 Orbital parameters, solar constant, and insolation anomalies

The Earth's orbital parameters (eccentricity, obliquity, and longitude of perihelion) are time evolving through the Holocene from 11.5 ka to the preindustrial period following Berger (1978), which affect the seasonal and latitudinal distribution and

magnitude of solar radiation received at the top of the atmosphere. Here, all orbital parameters are interpolated from the originally temporal resolution of 1,000 years to one year to produce continuous orbital forcings as model inputs. More specifically, the eccentricity declined through the Holocene, indicating a gradual decrease in the seasonal difference of incoming insolation since 11.5 ka. The obliquity was maximal at 9 ka, when the insolation difference was largest between low and high latitudes. The perihelion occurred close to the boreal summer solstice, autumnal equinox, and winter solstice at 11.5

ka, 6 ka, and the preindustrial period, respectively. Due to the above changes, summer (June–July–August) insolation averaged in the Northern Hemisphere slightly increased from 465 W m$^{-2}$ at 11.5 ka to a maximum of 468 W m$^{-2}$ at about 9.3 ka, and then continuously declined until the preindustrial period. June insolation at 60°N gradually decreased through the Holocene, whereas December insolation at 60°S increased since 11.5 ka and was maximal around 2 ka (Fig. 1a). Relative to the preindustrial period, June insolation anomalies generally increased with latitude and decreased with time, with positive

maximums of ~50 W m$^{-2}$ occurring at northern high latitudes during the early Holocene (Fig. 2).

The solar constant is fixed to 1365 W m$^{-2}$ throughout the run, which is in line with the PMIP3 experiment set up (PMIP PI Working Group, 2010). As mentioned in Otto-Bliesner et al. (2017), this value is higher than the value for the solar constant used by the models in PMIP4 (1360.7 W m$^{-2}$), which will lead to a global increase of incoming solar radiation compared to the PMIP4 experiments.

### 3.2.2 Atmospheric GHGs

During the Holocene, CFCs is fixed at 0 and $O_3$ is set to the PMIP3/CMIP5 preindustrial value as the early Holocene spin-up. Other atmospheric GHGs are time evolving through the Holocene and adjusted to the AICC2012 age model (Veres et al., 2013), with $CO_2$ following Bereiter et al. (2015), $CH_4$ following Loulergue et al. (2008) and $N_2O$ following Schilt et al. (2010).





The high-resolution atmospheric $CO_2$ concentrations are revised EPICA Dome C (EDC; Monnin et al., 2001, 2004) and
Antarctic composite ice core data built from the Law Dome (MacFarling Meure et al., 2006; Rubino et al., 2013) and West
Antarctic Ice Sheet Divide (Marcott et al., 2014), which show a slight decrease trend from 11.5 to 7 ka and an increase trend
afterward (Fig. 1b). The record of atmospheric $CH_4$ concentrations is built from the Antarctic EDC (Loulergue et al., 2008),
with a decreasing trend from the early- to mid-Holocene followed by a rising trend (Fig. 1c). The low-resolution atmospheric
concentrations of $N_2O$ derived from the Talos Dome ice core (Schilt et al., 2010) show large variabilities during the Holocene,
with relatively high values around 11 ka and during 6–0 ka but low levels between 11 and 6 ka (Fig. 1d). All above GHG
records are interpolated to the temporal resolution of one year to produce continuous forcings as model inputs.

Note that the chronologically accurate values of $CO_2$, $CH_4$, and $N_2O$ from the above reconstructions slightly differ from the
defined 6 ka concentrations in PMIP4. The nearest measured $CO_2$ concentration to 6 ka is 266.7 ppm from Bereiter et al.
(2015), which is higher than the 6 ka value of 264.4 ppm in PMIP4 derived from the original EDC ice core (Monnin et al.,
2001, 2004) after a smoothing spline (Otto-Bliesner et al., 2017). In contrast, both the nearest measured $CH_4$ and $N_2O$
concentrations of 586 ppb (Loulergue et al., 2008) and 252.4 ppb (Schilt et al., 2010) produced on the AICC2012 chronology
(Veres et al., 2013) are lower than the 6 ka values of 597 ppb and 262 ppb in PMIP4 (Otto-Bliesner et al., 2017), respectively,
on the original EDC1 chronology (Spahni et al., 2005).

### 3.2.3 Ice sheets and topography

The ice sheet extent and topography are time evolving from 11.5 ka to the preindustrial period using the ICE-6G_C
reconstruction (Argus et al., 2014; Peltier et al., 2015), which is based on Glacial Isostatic Adjustment modelling and
constrained by Global Positioning System measurements of vertical motion of the crust, exposure age dating of ice thicknesses,
relative sea level histories, and space-based gravity data by satellite system (Peltier et al., 2015). During the Holocene, the
most prominent change of ice sheets occurs in the Northern Hemisphere, especially in North America. According to the ICE-
6G_C reconstruction, the North American ice sheet retreated gradually from 11.5 ka to 9 ka, with the Laurentide ice sheet
holding ~12.5 m global mean sea level rise equivalent ice volume at 10 ka (Matero et al., 2020), retreated sharply from 9 ka
to 7 ka, and almost disappeared after 6 ka (Fig. 3) following the data provided by Dyke (2004). The Eurasian ice sheet retreated
since 11.5 ka and deglaciated after ~10 ka following Gyllencreutz et al. (2007), whereas the Greenland ice sheet was overall
stable from 11.5 ka to 9 ka and slightly retreated at its marginal regions during 8–7 ka (Fig. 3). Averaged over the Northern
Hemisphere, the ice area continuously declined from ~$1.2 \times 10^7$ km$^2$ at 11.5 ka to ~$0.2 \times 10^7$ km$^2$ during the mid-Holocene
(~7–5 ka), and remained little changed since then (Fig. 1e). In contrast, there is little variation in the Southern Hemispheric ice
sheets during the Holocene, with only a slight decrease in the ice extent from 11.5 ka to 10 ka around the Antarctica (Fig. 3).
The orography evolves to be consistent with the ice sheet extent, with the elevation of ice sheet below 2000 m, 2500 m, and
3500 m over Eurasia, North America, and Greenland, respectively, and reaching 4000 m over Antarctica through the Holocene
(Fig. 3). Besides, land surface properties are also adjusted for changes in ice extent. Where ice retreats, land surface is modified
to the prescribed vegetation at the preindustrial period.



The original ICE-6G_C data set is provided at 10 arcmin (~0.17°) horizontal resolution and at intervals of 500 years through the Holocene. Both included ice extent and topography data are interpolated to the time interval of 250 years, and updated every 250 years in the transient run. Considering that coastlines, bathymetry, and land–sea distributions are little varied during the Holocene, they are fixed as the preindustrial states in all runs. In addition, the meltwater flux change is not considered here. All other forcings, such as the vegetation, land surface and dust parameters, are prescribed at the preindustrial levels.

### 3.2.4 Experimental setup

Four transient experiments are run in total in the HT-11.5ka simulations, one all-forcing experiment and three single-forcing experiments. The orbital parameters, atmospheric GHGs, ice sheets and topography as detailed above are synchronously evolving with time in the all-forcing experiment (hereinafter HT-ALL), while the three single-forcing experiments are forced by only one of the above forcings, with the other two forcings fixed at 11.5 ka (hereinafter HT-ORB, HT-GHG, and HT-ICE, respectively) (Table 2). All four experiments are transiently run from 11.5 ka to the preindustrial period at 1850 CE for 11400 years. Monthly output data are provided for each simulation.

### 4 Preliminary simulation results

On the millennium timescale, the global annual mean surface (2 m) air temperature (GMST) in the HT-ALL simulation slightly decreases from 11.5 ka (~13.20 °C) to 7.5 ka (~12.93 °C), with an abrupt cooling at approximate 7.5 ka, and then continuously increases until the preindustrial period (13.57 °C) (Fig. 4). The value during the preindustrial period is very close to that of 13.61 °C from the median of the 14 PMIP4 models, although with a spread among the individual models ranging from 12.59 °C in IPSL-CM6A-LR to 14.98 °C in MIROC-ES2L. At 6 ka, the GMST is approximate 13.16 °C, which resembles the value of 13.28 °C from the mid-Holocene simulation in the 14-PMIP4-model median and is within the range from 12.20 °C (IPSL-CM6A-LR) to 14.53 °C (MIROC-ES2L) in the individual model simulations. Relative to the average for 1750–1850 CE, there is a global annual mean cooling of about 0.41 °C at 6 ka in HT-ALL simulation, which lies within the range of the results from the PMIP4 models. In the 14 PMIP4 models, the mid-Holocene cooling varied from 0.18 °C in FGOALS-g3 to 0.49 °C in FGOALS-f3-L with respect to the preindustrial period, with an average of 0.33 °C for the median of all models (Fig. 5). Note that the GMST simulated by CESM1.2.1 in HT-ALL simulation is lower than that by CESM2 in the PMIP4 both for 6 ka and preindustrial simulations (13.16 vs. 13.88 °C and 13.57 vs. 14.08 °C, respectively), but the mid-Holocene cooling magnitude for the former is nearly double that of the latter (0.41 vs. 0.20 °C). The warming trend since the mid-Holocene simulated by CESM1.2.1 in our HT-11.5ka transient experiment is similar to that in the CCSM3, LOVECLIM, and FAMOUS transient simulations (Liu et al., 2014), but opposite to the cooling trend as derived from the multi-proxy reconstructions (Marcott et al., 2013; Kaufman et al., 2020a), which is the so called "Holocene temperature conundrum". The abrupt cooling at approximate 7.5 ka is also seen in the LOVECLIM transient experiment (Liu et al., 2014). In addition, the GMST evolution from early- to mid-Holocene in our CESM1.2.1 transient simulation is somewhat different from the warming trend both in reconstructions





and other Holocene transient simulations (Marcott et al., 2013; Liu et al., 2014; Kaufman et al., 2020a), and the reasons for the discrepancies need for further investigations.

Fig. 5 further displays the GMST evolution since 10 ka in the all-forcing and single-forcing simulations. All time series are the anomalies relative to the last 100 years (1750–1850 CE) of the HT-ALL simulation. In response to the orbital forcing alone, the GMST slightly decreases during the past 10 ka, with an averaged cooling of approximate 0.2 °C in HT-ORB simulation. The GMST first decreases from 10 ka to 6.5 ka and then steadily increases until now in HT-GHG simulation, which closely follows the evolution of the atmospheric concentration of $CO_2$ and is also modulated by that of $CH_4$ and $N_2O$. In HT-ICE

simulation, the GMST increases from 10 ka to 6 ka in response to the retreat of ice sheets in the Northern Hemisphere, and then slightly decreases possibly due to the lag effect of ice sheet melting. Taken together, the weak cooling trend during the early Holocene in HT-ALL simulation is mainly driven by both decreased orbital insolation and GHG concentrations, while the retreat of ice sheets plays an opposite role. On the other hand, the strong warming trend since 7.5 ka is mainly induced by increased concentrations of GHGs, which is largely resulted from that of $CO_2$, with a simultaneously positive contribution

from the retreat of ice sheets during 7.5–6 ka and a negative contribution from the orbital insolation throughout the past 7.5 ka.

## 5 Summary

The Holocene presents a host of opportunities to study the climate evolution over the latest interglacial period including the mean climate trend on the millennium scale and abrupt changes on the time scales from decades to centuries. The climate

model provides a useful tool to investigate the underlying dynamic mechanism for the climate change during this well-studied time period, in particular the improvements of the model and computational power make it possible to run transient simulations to comprehensively understand the climate evolution in response to time-varying external forcings and feedbacks. Several modelling studies have begun this work, but most of them are performed with climate system models or Earth system models with low spatial resolutions and/or acceleration techniques to reduce computational resources. Therefore, we have carried out

a new set of unaccelerated transient experiments covering the whole Holocene since 11.5 ka (HT-11.5ka) using a comprehensive Earth system model with relatively high spatial resolutions, considering both the full-forcing and single-forcing effects of the main boundary conditions.

The HT-11.5ka simulations are run with the NCAR CESM1.2.1 with an atmospheric resolution of approximately $1.9° \times 2.5°$ in the horizontal and 26 layers in the vertical and an oceanic resolution of approximately 1° and 60 levels in the horizontal and

vertical respectively. All simulations are spin up from the early Holocene at 11.5 ka and transiently run from 11.5 ka to the preindustrial period at 1850 CE with boundary conditions of Earth's orbital parameters, atmospheric GHGs, and ice sheets and orography that have been provided on the PMIP4 wiki for the transient deglaciation (PMIP Last Deglaciation Working Group, 2016; https://pmip4.lsce.ipsl.fr/doku.php/exp_design:degla). Four transient experiments are run in total, one all-forcing experiment forced by synchronously time-evolving orbital parameters, atmospheric GHGs, and ice sheets and three single-



forcing experiments forced by only one of the above forcings with the other two forcings fixed at 11.5 ka. The all-forcing simulation result shows a slight decrease of the GMST from 11.5 ka to 7.5 ka due to both decreases in orbital insolation and GHG concentrations, with an abrupt cooling at approximate 7.5 ka, which is followed by a continuous warming until the preindustrial period mainly resulted from increased GHG concentrations. Relative to the preindustrial period, the simulated cooling magnitude in the GMST at 6 ka is close to that from the median of the 14 PMIP4 model simulations and within the range of values in the individual model simulations. The late Holocene warming trend simulated by CESM1.2.1 in our all-forcing simulation is consistent with that in the CCSM3, LOVECLIM, and FAMOUS transient simulations as shown in Liu et al. (2014), but inconsistent with the reconstructed cooling trend in Marcott et al. (2013) and Kaufman et al. (2020a).

The present HT-11.5ka transient simulations for the whole Holocene, spanning from 11.5 ka to the preindustrial period, are designed to investigate both combined and separated effects of the main external forcing of orbital insolation, atmospheric GHGs, and ice sheets on the climate evolution over the Holocene, which will be the supplement for the PMIP4 transient Core experiments for the last deglaciation from 21 to 9 ka (Ivanovic et al., 2016) and the PMIP4 Tier 3 transient simulations for the Holocene from 6 to 0 ka (Otto-Bliesner et al., 2017). Although these new set of transient Holocene simulations are run with a high-resolution version of the comprehensive Earth system model, the meltwater flux change, solar irradiance and volcanic forcing, land-use and vegetation changes, dust and aerosol effects, and stratospheric chemistry and dynamics are not fully considered here. As a first step, this study provides the detailed model description, experimental design, boundary conditions, and some preliminary results for the new set of HT-11.5ka transient simulations. Further analyses on the simulation results including the mean climate evolution and abrupt climate changes over the Holocene both at global and regional scales, the underlying dynamic mechanisms, and comparisons with other transient simulations covering the Holocene and multi-proxy-based reconstructions, will be carried out by a series of follow-up studies.

## 6 Code and data availability

All boundary condition data required for running the HT-11.5ka experiments (summarized in Tables 1 and 2) can be downloaded from the PMIP4 last deglaciation wiki (PMIP Last Deglaciation Working Group, 2016; https://pmip4.lsce.ipsl.fr/doku.php/exp_design:degla). The CESM1.2.1 source code is available on the CESM official website (https://svn-ccsm-models.cgd.ucar.edu/cesm1/release_tags/cesm1_2_1). The PMIP4 model output for the mid-Holocene and preindustrial experiments can be downloaded via the Earth System Grid Federation (ESGF; https://esgf-node.llnl.gov/search/cmip6/). The GMST data from the HT-11.5ka transient simulation shown in this study can be accessed at https://zenodo.org/record/6269566#.Yhg79OhBxLo (last access: 25 February 2022) and is archived at https://doi.org/10.5281/zenodo.6269566. Other HT-11.5ka transient simulation data are available upon reasonable request to the first author and can also be reproduced with the above code and boundary condition data.



*Author contributions.* D. Jiang conceived the idea and designed the HT-11.5ka experiments. B. Su collected all boundary condition data for the simulations, debugged the CESM1.2.1 model, and run the early Holocene spin-up simulation. B. Su and R. Zhang collaborated to perform the all-forcing and three single-forcing transient simulations. Z. Tian wrote the manuscript, processed the output data from all simulations, and produced the figures with contributions from all authors.

*Competing interests*. The authors declare that they have no conflict of interest.

*Acknowledgements*. We are grateful to the NCAR CESM and PMIP4 climate modeling groups for producing and sharing their model source code and model output; to Jean-Yves Peterschmitt (Laboratoire des Sciences du Climat et de l'Environnement, France) for managing and archiving the boundary conditions for the transient deglaciation experiment, as well as setting up and maintaining the PMIP4 last deglaciation wiki pages. This research was supported by the National Key R&D Program of China (2017YFA0603404), the Strategic Priority Research Program of Chinese Academy of Sciences (XDA20070103), and

the National Natural Science Foundation of China (41931181, 42075048).

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





**Table 1.** Summary of model boundary conditions for the early Holocene (11.5 ka) spin-up and Holocene transient (11.5 ka to 1850 CE) experiments; see text for details. Data are available from PMIP4 last deglaciation wiki (PMIP Last Deglaciation Working Group, 2016). Boundary condition group headings are in bold.

| Experiment type | Boundary conditions | Description |
|---|---|---|
| Early-Holocene (11.5 ka) spin-up | **Orbital parameters** | |
| | Eccentricity | 0.01957212 |
| | Obliquity | 24.17958° |
| | Perihelion–180° | 270.2132° |
| | Date of vernal equinox | Noon on 21 March |
| | **Solar constant** | 1365 W m$^{-2}$ |
| | **GHGs** | |
| | Carbon dioxide ($CO_2$) | 267.4 ppm |
| | Methane ($CH_4$) | 652.4 ppb |
| | Nitrous oxide ($N_2O$) | 251.1 ppb |
| | Chlorofluorocarbon (CFC) | 0 |
| | Ozone ($O_3$) | Preindustrial (1850 CE) |
| | **Ice sheets and orography** | 11.5 ka data from ICE-6G_C reconstruction (Argus et al., 2014; Peltier et al., 2015) |
| | **All others** | Prescribed preindustrial (1850 CE) levels |
| Holocene Transient (11.5 ka to 1850 CE; HT-11.5ka) | **Orbital parameters** | All orbital parameters should be transient as per Berger (1978) |
| | **GHGs** | Adjusted to the AICC2012 chronology (Veres et al., 2013) |
| | Carbon dioxide ($CO_2$) | Transient, as per Bereiter et al. (2015) |
| | Methane ($CH_4$) | Transient, as per Loulergue et al. (2008) |
| | Nitrous oxide ($N_2O$) | Transient, as per Schilt et al. (2010) |
| | **Ice sheets and orography** | Transient with ICE-6G_C reconstruction, updated every 250 years |
| | **All others** | As per early Holocene (11.5 ka) spin-up experiment |





**Table 2.** Boundary conditions for HT-11.5ka experiments.

|  | Orbital parameters | GHGs | Ice sheets | Time span |
| --- | --- | --- | --- | --- |
| HT-ALL | Transient | Transient | Transient | 11.5 ka to 1850 CE |
| HT-ORB | Transient | Fixed at 11.5 ka | Fixed at 11.5 ka | 11.5 ka to 1850 CE |
| HT-GHG | Fixed at 11.5 ka | Transient | Fixed at 11.5 ka | 11.5 ka to 1850 CE |
| HT-ICE | Fixed at 11.5 ka | Fixed at 11.5 ka | Transient | 11.5 ka to 1850 CE |


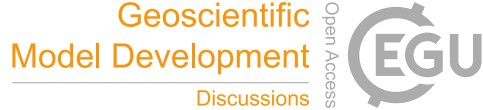



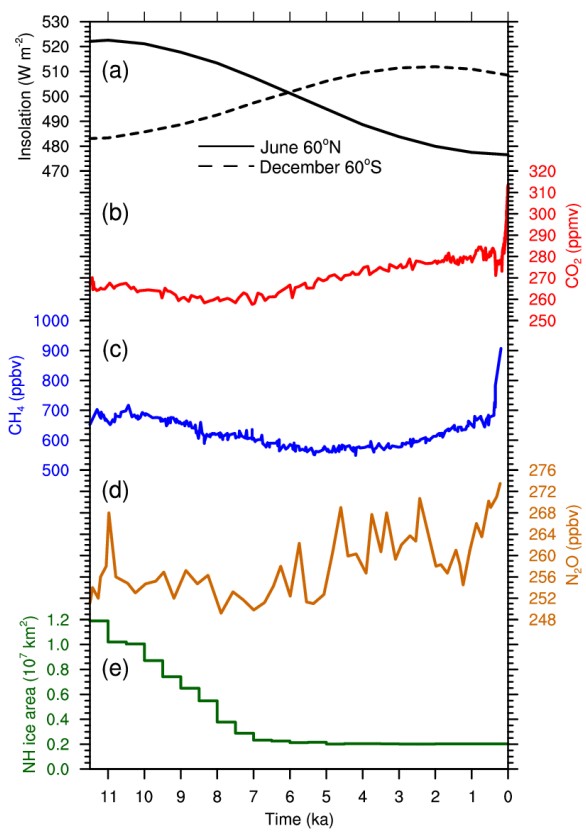

**Figure 1: Temporal evolution of the external forcings in HT-11.5ka simulations since the early Holocene (11.5 ka) for (a) June insolation at 60°N (solid line) and December insolation at 60°S (dashed line) (Berger, 1978), (b) atmospheric CO₂ concentration (units: ppm) (recent composite of EPICA Dome C, Law Dome, and West Antarctic Ice Sheet Divide records, Antarctica; Bereiter et al., 2015), (c) atmospheric CH₄ concentration (EPICA Dome C, Antarctica; Loulergue et al., 2008), (d) Atmospheric N₂O**
**concentration (Talos Dome, Antarctica; Schilt et al., 2010), and (e) area of the ice sheets in the Northern Hemisphere according to the ICE-6G_C reconstruction (Argus et al., 2014; Peltier et al., 2015).**



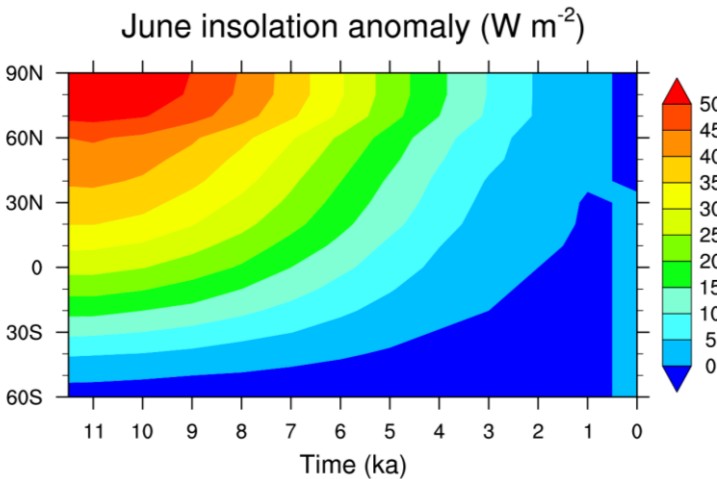

**Figure 2: Latitude–time anomalies in June insolation relative to the average for the last 1000 years according to Berger (1978).**



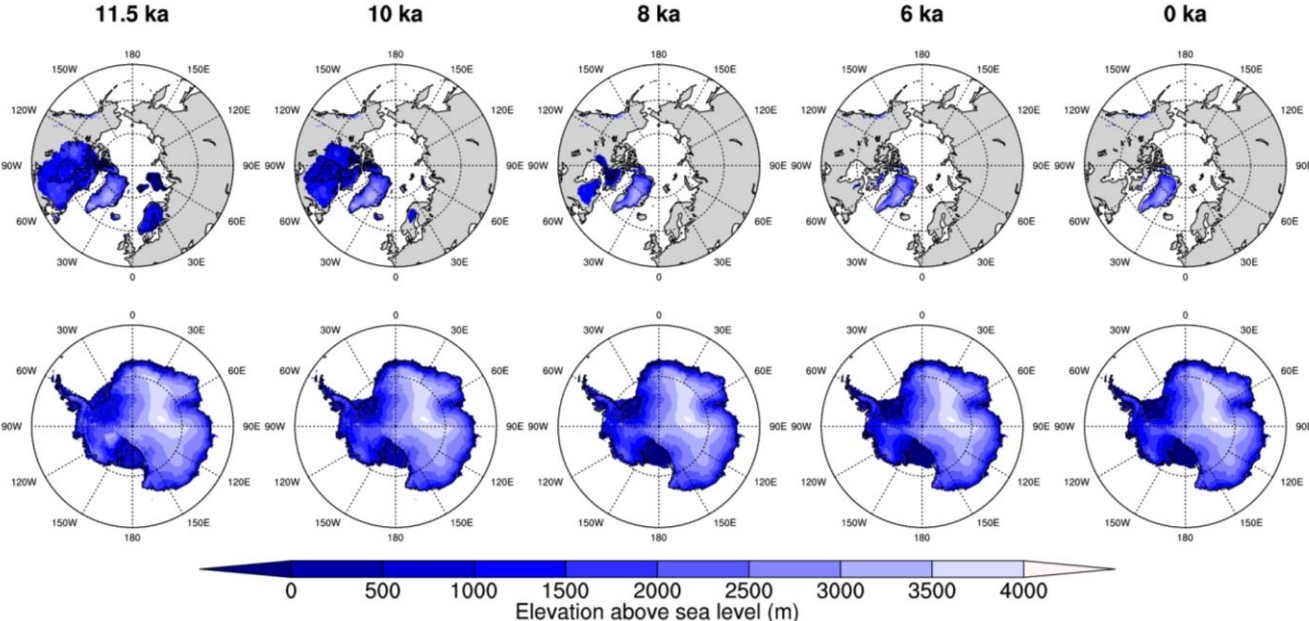

**Figure 3: Northern (top) and Southern Hemisphere (bottom) ice sheet elevation at 11.5, 10, 8, 6 and 0 ka (1950 CE) for the ICE-6G_C reconstruction at 10 arcmin horizontal resolution; elevation is plotted where the fractional ice mask is more than 0.5 (Peltier et al., 2015).**

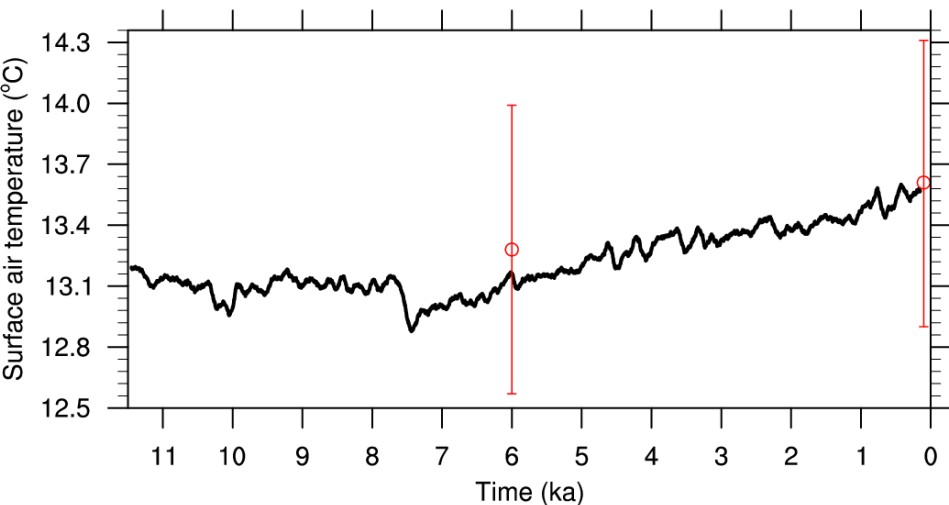

**Figure 4: Global annual mean surface air temperature since 11.5 ka under the all-forcing simulation. The time series is smoothed**
**by a 101-year moving average. The red circles at 6 ka and 0.1 ka stand for the mid-Holocene and preindustrial results simulated by**
**the median of the 14 PMIP4 models, with red vertical error bars representing plus and minus one standard deviation for 14 models.**





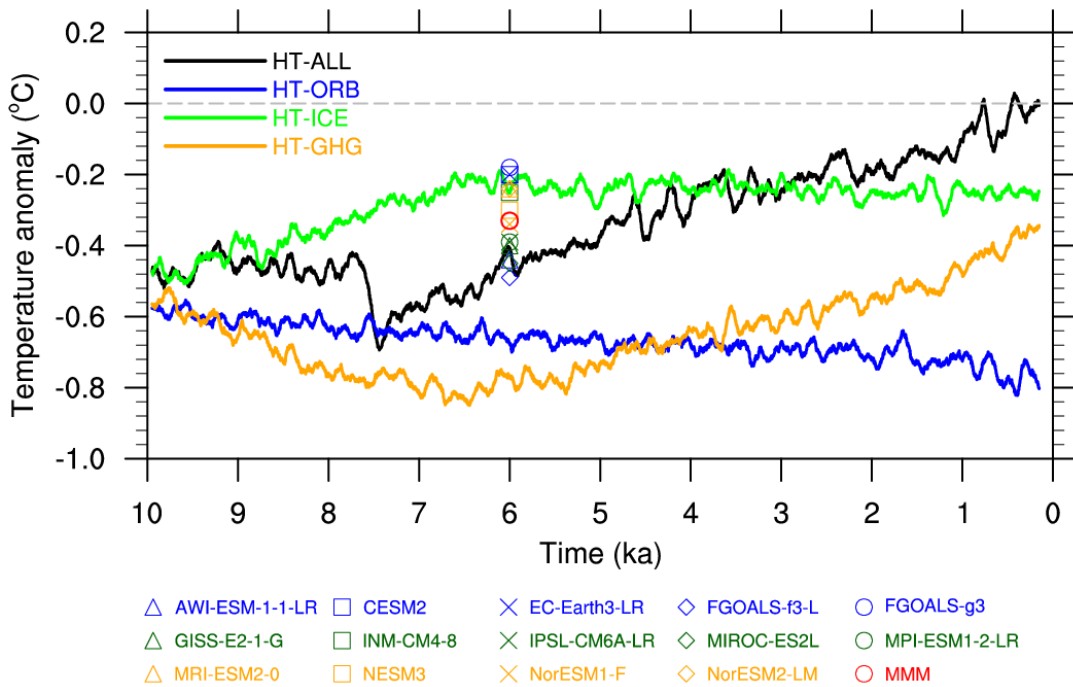

**Figure 5: Global annual mean surface air temperature anomaly since 10 ka under the all-forcing (HT-ALL, black) and the single-forcing (HT-ORB, blue; HT-ICE, green; and HT-GHG, orange) simulations relative to the last 100 years of the all-forcing simulation.**
**All of the time series are smoothed by a 101-year moving average. The triangle, square, cross, diamond, and circle symbols at 6 ka stand for the mid-Holocene minus preindustrial results simulated by the 14 PMIP4 models and their multi-model median (MMM).**