# Peer review of "Transient climate simulations of the Holocene (version 1) – experimental design and boundary conditions"

_Geoscientific Model Development, 2022_

## Author Response (AR1)

**Response to Comments from Referees**

We thank the referees for insightful and constructive comments. Below is our point-to-point reply to these comments (the referee's comments are in blue, and our responses are in black).

The paper describes the experimental design of a small set of transient Holocene simulations, and a few results.

Part 1.1 is a honest summary of the present knowledge of the climate evolution over the Holocene. Part 1.2 is a honest summary of the main results from previous simulations of the Holocene by different groups. Both parts are based on a good review of the present literature, and I appreciated this synthesis. They provide a good base for part 1.3 which explains clearly why new Holocene simulations, with ESMs (not EMICs), unaccelerated models, etc ... might bring new insights about the Holocene climate.

Part 2 is a very short model description. There is no need to get more in depth, as the model is fully described in the cited literature.

Part 3 describes the experimental design. The description is comprehensive, and would allow any modeller to run a similar set of experiments with its own model (or with the same model).

Part 4 gives a few preliminary results.

The paper is clear, easy to read, with a relevant structure and progression. Abstract and conclusion are fully supported by the main text. The langage seems good, but my own english does not allow me to have a relevant evaluation. I didn't find any typo. Figures are clear and readable. I thank the authors for the care taken with the manuscript.

I have no major concern about the paper, which perfectly fits the GMD category "Model experiment description paper".

I have just one minor concern.

Line 233 reads that the spin-up run has a 'stable' Atlantic Meridional Overturning Circulation (AMOC). Most (all ?) models show some variability of the AMOC on inter annual to decadal time scales. I would appreciate to get an idea of the amplitude of the inter annual to decadal variability of this experiment, particularly for the AMOC. If the amplitude is strong, the choice to start the transient experiment form a state with high or low AMOC might have some impact on the result.

We agree that models generally show some variability of the AMOC on inter annual to decadal time scales. We have drawn the time series of the AMOC strength (defined as the maximum of the mean meridional mass overturning streamfunction below 500 m north of 28°N in the Atlantic) for the 1500 years in the early Holocene spin-up experiment. Although it shows some variability on inter annual to decadal time scales, the AMOC strength is relatively stable for the last 100 years with an amplitude of approximately 25.4 Sv. This AMOC amplitude is reasonable, since it lies within the range of the 10 PMIP4 models and is comparable to the amplitude in GISS-E2-1-G and FGOAL-f3-L for the preindustrial experiments as shown in Brierley et al. (2020). The stable trends of the AMOC amplitude as well as global mean surface air and sea surface temperatures for the last 100 years in the spin-up experiment mean that the model has reached a quasi-equilibrium state as suggested by Kageyama et al. (2018). Therefore,

it is reasonable to start the transient experiment from this quasi-equilibrium state. We have added

the amplitude of the AMOC in the revised manuscript accordingly (Line 216).

References:

Brierley, C. M., Zhao, A., Harrison, S. P., et al.: Large-scale features and evaluation of the

PMIP4-CMIP6 midHolocene simulations, Clim. Past, 16, 1847–1872, doi:10.5194/cp-16-

1847-2020, 2020.

Kageyama, M., Braconnot, P., Harrison, S. P., et al.: The PMIP4 contribution to CMIP6 – Part

1: Overview and over-arching analysis plan, Geosci. Model Dev., 11, 1033–1057,

doi:10.5194/gmd-11-1033-2018, 2018.

Anonymous Referee #2

Review of the paper by Zhiping Tian et al. on Transient climate simulations of the Holocene (version 1) – experimental design and boundary conditions. The paper describes a model configuration for Holocene climate using CESM1.2.1. Compared to other PMIP participants it exploits a surprisingly high resolution of ~2° in the atmosphere and 1° in the ocean and does not employ any acceleration techniques. Using this setup the authors conducted some transient runs from 11.5ka until the pre-industrial era.

I like the paper in general but feel that the authors need to add more substantial material before it can be accepted by GMD. Hence I recommend the authors to resubmit the paper after significant revision.

1. The introduction leaves an impression that the authors know their topic and what they are doing. However, it consists of 3 parts and takes more than 30% of the paper. I see the authors need to make it more concise and keep only the relevant for the motivation parts.

   According to your suggestion, we have revised the introduction to make it more concise and keep only the relevant for the motivation parts. Please refer to the revised version for details.

2. I agree with the first reviewer that the model description lacks some details. Which parameterisations are used in the ocean and atmosphere, which coefficients etc. all this is not there. The presented setup does not really follow the PMIP4 protocol, doesn't it? I also wonder how the ocean was initialized for the spinup?

   Compared to previous model versions, the physical improvements in the oceanic component of POP2, used in both CESM1.2.1 and CCSM4, include a near-surface eddy flux parameterization, an abyssal tidally mixing parameterization, an overflow parameterization to represent the Denmark Strait, Faroe Bank Channel, Weddell Sea, and Ross Sea overflows, a submesoscale mixing scheme, vertically-varying thickness and isopycnal diffusivity coefficients,

and modified and generally reduced horizontal viscosity coefficients as detailed in Danabasoglu et al. (2012) and Hurrell et al. (2013). The atmospheric component of CAM4 contains notable improvements in the deep convection parameterization, in which the calculation of Convective Available Potential Energy (CAPE) has been reformulated to include more realistic dilution effects through an explicit representation of entrainment, and sub-grid scale convective momentum transports have also been added. The CAM4 also contains improvements in the cloud fraction parameterization with a freeze-drying process contributing to a greater consistency between polar cloud fraction and water condensate properties, and in the radiation interface and computational scalability as detailed in Neale et al. (2010). The above details in the model description on the parameterizations and coefficients used in the ocean and atmosphere have been added in the revised manuscript accordingly (Lines 168–172 and 176–180).

Most of the presented setup for our all-forcing transient experiment follows the PMIP4 protocol for the transient deglaciation (21–0 ka) experiments, including the boundary conditions of astronomical parameters, trace gases, ice sheet and orography from 11.5 ka to the preindustrial period. The meltwater flux change is not considered here, which is also one of the three options in the PMIP4 last deglacial experiment protocol (Lines 278–279). Since our transient simulations start from the early Holocene at 11.5 ka, the spin-up experimental setup is different from that recommended for PMIP4 transient deglaciation experiments, the latter of which uses the last glacial maximum spinup (21 ka) or the transient orbit and trace gases spinup (26–21 ka). In particular, the initial ocean condition is taken from an archived 500-yr spinup from NCAR model case "b40.1850.track1.2deg.003", a preindustrial control experiment performed by CCSM4 with $2°$ resolution in the atmosphere and $1°$ resolution in the ocean, which can be downloaded from the CCSM4 input data archive (https://svn-ccsminputdata.cgd.ucar.edu/trunk/inputdata/ccsm4_init/b40.1850.track1.2deg.003/0501-01-01/). We

have added this information in the revised version (Lines 210–212 and 404–407).

3. line 295: Is it a standard practice in PMIP? I expect the effect is enorm!

There are three options for the freshwater fluxes provided for the PMIP4 transient

deglaciation experiments: melt-uniform, melt-routed and no-melt

(https://pmip4.lsce.ipsl.fr/doku.php/exp_design:degla). As one of the above options, the

meltwater flux change is not considered in our Holocene transient simulations. Considering the

high computational cost with the relatively high-resolution model (2° resolution in the

atmosphere and 1° resolution in the ocean), the main purpose of running this set of Holocene

transient simulations is to investigate the full- and single-forcing effects of several most

important boundary conditions, including the orbital parameters, GHG concentrations, and ice

sheets, on the Holocene climate evolution. The meltwater flux change, solar irradiance and

volcanic forcing, land-use and vegetation changes, dust and aerosol effects, and stratospheric

chemistry and dynamics are not fully considered here, all of which will have effects on the results

as you pointed out. We have added the statements in the revised manuscript accordingly (Lines

278–279 and 391–394).

4. For a GMD paper on the model setup description it would be necessary to present the
   model performance, scalability and throughput for different model components
   distinguishing between IO, ocean and atmosphere costs etc.

Since this is a GMD paper on the model experiment description rather than on the model

description, our main focus is on the model experimental setup and boundary conditions for the

Holocene transient climate simulations, which is detailed in Section 3. Here we only give a very

short model description in Section 2. Both us and the first reviewer think that there is no need to

get more in depth, as the model is fully described in the cited literature (Lines 165–183). As you

suggested before, more details in the model description on the parameterizations and coefficients

used in the ocean and atmosphere have been added in the revised manuscript accordingly (Lines

168–172 and 176–180). Further information on the model performance, scalability, and

throughput for different model components distinguishing between IO, ocean and atmosphere

costs, etc., as you kindly mentioned above, are documented in the CCSM4/CESM1 special

collection of the *Journal of Climate* (see http://journals.ametsoc.org/collection/CCSM4-

CESM1), and additional model simulations for the past and present climates can be found, for

instance, in Goldner et al. (2014), Song and Zhang (2018, 2019), and Park et al. (2019). We have

added this information in the revised version (Lines 183–187).

5. My main comment is that the preliminary results are limited by one page and one plot
   which sounds a bit poor (10% of the paper). In the discussion of Fig.5 the authors try to
   interpret the drivers of Holocene climate GMST anomaly by using the additional 4
   experiments where they sequentially turn off different boundary conditions. If I interpret
   Fig. 5 correctly I see that the anomalies in these 4 experiments do not sum to the
   reference experiment (HT-ALL) containing all boundary conditions. This points to the
   high level of nonlinearity in the system and reduces the confidence of the discussion in
   the results section.

    In the revised manuscript, we have added more preliminary results about the spatial

distribution and zonal average of annual and seasonal surface air temperature changes at 6 ka in

the HT-ALL simulation, as well as their comparisons with the mid-Holocene changes in the 14

PMIP4 model simulations and their arithmetic mean. Section 4 now extends to more than two

pages with the addition of one Table and two plots. Please refer to the revised Section 4 (Lines

299–343), Table 3, and Figures 6 and 7 for more details.

You are right that relative to the average for 1750–1850 CE (0.2–0.1 ka) of the full-forcing (HT-ALL) simulation, the annual GMST anomalies since 10 ka in the three single-forcing experiments (HT-ORB, HT-GHG, and HT-ICE) do not sum to the anomaly in the reference experiment (HT-ALL) containing all boundary conditions as seen from Fig. 5 or the following Fig. R1D, which points to a level of nonlinearity in the system. First of all, rather than sequentially turning off different boundary conditions, the three single-forcing experiments are forced by only one of the three boundary conditions (i.e., orbital parameters, atmospheric GHGs, and ice sheets), with the other two boundary conditions fixed at 11.5 ka (see Lines 282–287 and Table 2 for experimental setup) in this study. Second, this nonlinearity in the system revealed from our HT-11.5 ka transient simulations performed by CESM1.2.1 also exists in other transient simulations covering the whole Holocene performed by CCSM3, LOVECLIM, and FOMOUS as shown in Liu et al. (2014). As displayed in the following Fig. R1A–C, relative to the average values between 1.5 and 0.5 ka of the full-forcing simulation, the global annual mean temperature anomalies in the single-forcing experiments do not simply sum to the anomaly in the all-forcing experiment in CCSM3, LOVECLIM, and FOMOUS, which is particularly evident for the evolution since 10 ka. However, after simply summing the anomalies in the single-forcing experiments and then through some translations or parallel shifts, the gray curve in Fig. R1A–C (the "sum" of the single-forcing simulations) is overall comparable to the all-forcing simulation in the three models. This kind of summation, through some translations or parallel shifts, also reflects a level of nonlinearity in the system. Therefore, the nonlinearity in the system is not unique in our transient simulations performed by CESM1.2.1, but is commonly existed in other transient simulations performed by CCSM3, LOVECLIM, and FOMOUS, which will have some

effects on the results and calls for further investigation in the future. We have added this point in

the revised manuscript accordingly (Lines 355–359).

[Figure]

Figure R1: Global annual mean temperature anomaly under the all forcing (ALL, black) and the

single forcings [GHG, orange; orbital, cyan; ICE, green; and meltwater flux, magenta (only in

CCSM3)], as well as the sum of the single forcing simulations (SUM, gray) relative to the average

values between 1.5 and 0.5 ka of the full-forcing simulation in (A) CCSM3, (B) LOVECLIM,

and (C) FAMOUS as adopted from Liu et al. (2014). (D) Global annual mean temperature

anomaly since 10 ka under the all-forcing (HT-ALL, black) and the single-forcing (HT-ORB,

blue; HT-ICE, green; and HT-GHG, orange) simulations relative to the last 101 years (0.2–0.1

ka) of the full-forcing (HT-ALL) simulation by CESM1.2.1 from this study.

Reference:

Liu, Z., Zhu, J., Rosenthal, Y., Zhang, X., Otto-Bliesner, B. L., Timmermann, A., Smith, R. S., Lohmann, G., Zheng, W., and Timm, O. E.: The Holocene temperature conundrum, P. Natl. Acad. Sci., 111, E3501–E3505, doi:10.1073/pnas.1407229111, 2014.

6. I would like to see some key diagnostics from PMIP4 and how the HT-ALL experiment aligns with the other models. As an example one could look into Chris M. Brierley 2020 (https://doi.org/10.5194/cp-16-1847-2020, 2020) and do the comparison.

As you suggested, we have added some key diagnostics from PMIP4 and how the HT-ALL experiment aligns with the other models in the revised Section 4 with the additions of Table 3 and Figures 6 and 7. More specifically, following Brierley et al. (2020) as you mentioned, the spatial distribution of annual and seasonal (DJF and JJA mean) surface air temperature changes at 6 ka in the HT-ALL simulation is additionally displayed in Figure 6 and compared with the mid-Holocene changes in the arithmetic mean of the 14 PMIP4 models. Moreover, the global and zonal mean changes in annual and seasonal temperature changes at 6 ka averaged over the zonal bands of 60°–90°N, 30°–60°N, 0°–30°N, 30°S–0°, 60°–30°S, and 90°–60°S in the HT-ALL simulation are quantitatively compared with those in the 14 PMIP4 models and their arithmetic mean (Table 3 and Figure 7). As a whole, the above key diagnostics show that both at global and zonal mean scales, the annual and seasonal mean changes in surface air temperature at 6 ka in our HT-ALL simulation lie within the range of the 14 PMIP4 model results and are overall stronger than their arithmetic means. In particular, the annual and seasonal mean GMSTs simulated by CESM1.2.1 in the HT-ALL simulation are lower than those by CESM2 in the PMIP4 both for 6 ka and preindustrial simulations, but the mid-Holocene annual and DJF cooling magnitudes for the former are nearly double those of the latter, and the JJA cooling in the former is opposite to and four times the JJA warming in the latter. There are also some differences,

mainly in magnitudes, for the zonal mean changes in annual and seasonal temperatures at 6 ka between the two versions of CESM. Please refer to the revised version for further details (Lines 299–343).

As a first step, this study provides the detailed model description, experimental design, boundary conditions, and some preliminary results for the new set of HT-11.5ka transient simulations. Further analyses on the simulation results including the mean climate evolution and abrupt climate changes over the Holocene both at global and regional scales, the underlying dynamic mechanisms, and comparisons with other transient simulations covering the Holocene and multi-proxy-based reconstructions, will be carried out by a series of follow-up studies.

---

## Author Response (AR2)

**Response to Comments from Referee #1**

We thank the referee for insightful and constructive comments. Below is our point-to-point reply to these comments (the reviewer's comments are in blue, and our responses are in black).

Review of

"Transient climate simulations of the Holocene (version 1) – experimental design and boundary conditions"
by Zhiping Tian, Dabang Jiang, Ran Zhang, Baohuang Su
for Geoscientific Model Development

The paper describes the experimental design of a small set of transient Holocene simulations, and a few results.

Part 1.1 is a honest summary of the present knowledge of the climate evolution over the Holocene. Part 1.2 is a honest summary of the main results from previous simulations of the Holocene by different groups. Both parts are based on a good review of the present literature, and I appreciated this synthesis. They provide a good base for part 1.3 which explains clearly why new Holocene simulations, with ESMs (not EMICs), unaccelerated models, etc ... might bring new insights about the Holocene climate.

Part 2 is a very short model description. There is no need to get more in depth, as the model is fully described in the cited literature.

Part 3 describes the experimental design. The description is comprehensive, and would allow any modeller to run a similar set of experiments with its own model (or with the same model).

Part 4 gives a few preliminary results.

In general, the paper is clear, easy to read, with a relevant structure and progression. Part 4 has been extended and some parts are now a bit hard to read. In this part, some

phrases contain a lot of figures, and it is sometimes hard to match up the figures and their region or season. Starting line 327 is a 5 lines sentence that contains 14 figures that describe 2 seasons and 3 latitude bands. Starting line 337 is a 4 lines sentences with figures. This part deserve some rephrasing. Maybe more tables could help the reader.

We have rephrased this part accordingly, and it is now much easier to read. The 5 lines sentence staring line 327 has now revised to three sentences (please refer to lines 327–331), and the 4 lines sentence starting the original line 337 has now revised to two sentences (please refer to lines 338–342). Since Table 3 has already given the global annual and seasonal mean temperatures in the simulations of HT-ALL and the 14 PMIP4 models, and Fig. 7 has further shown the corresponding zonal averaged changes, no more tables are given in the revised manuscript.

Abstract and conclusion are fully supported by the main text. The langage seems good, but my own english does not allow me to have a relevant evaluation. I didn't find any typo. Figures are clear and readable. I thank the authors for the care taken with the manuscript.

I have no major concern about the paper, which perfectly fits the GMD category "Model experiment description paper".